# Foaming and Physicochemical Properties of Commercial Protein Ingredients Used for Infant Formula Formulation

**DOI:** 10.3390/foods11223710

**Published:** 2022-11-18

**Authors:** Siyu Zhang, Jianjun Cheng, Qinggang Xie, Shilong Jiang, Yuxue Sun

**Affiliations:** 1College of Food Science, Northeast Agricultural University, Harbin 150030, China; 2Heilongjiang Feihe Dairy Co., Ltd., Beijing 100015, China

**Keywords:** infant formula, protein, foaming properties, physicochemical properties

## Abstract

Protein, as one of the main ingredients for infant formula, may be closely related to the undesirable foam formed during the reconstitution of infant formula. Demineralized whey powder (D70 and D90), whey protein concentrate (WPC), and skimmed milk powder (SMP) are the four protein ingredients commonly used in infant formula formulation. The foaming and physicochemical properties of these four protein ingredients from different manufacturers were analyzed in the present study. Significant differences (*p* < 0.05) in foaming properties were found between the samples from different manufacturers. SMP showed a highest foaming capacity (FC) and foam stability (FS), followed by D70, D90, and WPC. Although the protein composition was similar based on reducing SDS-PAGE, the aggregates varied based on non-reducing SDS-PAGE, probably resulting in the different foaming properties. Particle size, zeta potential, and solubility of the protein ingredients were assessed. The protein structure was evaluated by circular dichroism, surface hydrophobicity, and free sulfhydryl. Pearson’s correlation analysis demonstrated that FC and FS were positively correlated with random coil (0.55 and 0.74), *β*-turn (0.53 and 0.73), and zeta potential (0.55 and 0.51) but negatively correlated with *β*-strand (−0.56 and −0.71), free sulfhydryl (−0.56 and −0.63), particle size (−0.45 and −0.53), and fat content (−0.50 and −0.49). The results of this study could provide a theoretical guidance for reducing formation of foam of infant formula products during reconstitution.

## 1. Introduction

Infant formula is a complex mixture of proteins, lipids, carbohydrates, vitamins, minerals, etc. It plays an important role in bottle-fed infant growth owing to providing essential nutrients and certain bioactive components [1]. However, undesirable foam formed during the reconstitution may reduce the functionality of infant formula [2]. Foaming properties are an essential consideration in infant formula formulation.

As a surfactant, protein contains hydrophilic and hydrophobic groups, which play an important role in the undesirable foaming in infant formula [3,4]. The unfolding of the protein structure and the diffusion at the air–water interface are related to the foaming capacity (FC) and the viscoelastic film formed around the bubbles is associated with the foam stability (FS) [5]. Demineralized whey powder (DWP), whey protein concentrate (WPC), and skimmed milk powder (SMP) are the main protein ingredients used in infant formula formulation [6]. DWP is divided into various categories according to the degree of ash removal. Among them, D70 and D90 refer to DWP with ash removal degrees of 70% and 90% respectively, which are commonly used protein ingredients in infant formula formulation.

As complex mixtures, different compositions and physical states are found in D70, D90, WPC, and SMP ingredients [7]. D70 and D90 are complex mixtures of lactose, protein, fat, and small amounts of ash, in which lactose is the most abundant. In addition to a high content of protein, WPC also shows a higher fat content in comparison to the other three protein ingredients. For SMP, it contains whey protein and casein, in which casein is the most abundant. In addition, compared to D70, D90, and WPC, SMP exhibits the lowest content of fat, resulting from the separation of fat during processing. In the preparation of D70, D90, and WPC, the liquid whey from cheese making is pasteurized and separated for fat, ultrafiltered, filtered, evaporated, and then spray dried. Additionally, during the production of D70 and D90, electrodialysis and ion exchange are used to remove ash for reducing the kidney burden on the infant [7]. For the SMP, the raw material is milk rather than whey. 

The physicochemical properties of protein are influenced by different protein composition and secondary and conformational structure [8]. Various processing (e.g., fat separation, pasteurization, evaporative concentration, and spray drying) further affects the heat stability and structure of protein [9]. Thus, there may be differences in the foaming properties of commercial protein ingredients during infant formula production, which is attributed to the differences in the properties of the raw materials (milk quality, type of cheese), handling practices, and processing equipment [10]. Harper and Lee (1997) indicated that the standard WPC samples from six U.S. manufacturers significantly differed in color, particle size, and foaming properties [11]. In another study, functional properties and compositional parameters of 35 commercial SMP samples from six manufacturers were evaluated, demonstrating that similar properties were shown in ingredients from the same manufacturers compared to those of different manufacturers [12]. However, information on the foaming properties of these four substantial ingredients in the infant formula industry is not fully understood. 

Foaming properties are essential indicators for evaluating the reconstitution qualities of infant formula, which are affected by the type and source of protein ingredients. This study characterized and compared the foaming and physicochemical properties in 20 samples of D70, D90, WPC, and SMP from five manufacturers. The relationship between foaming properties and physicochemical properties was analyzed by Pearson correlation analysis. The potential indicator was further selected to distinguish the foaming properties of protein ingredients. These results provide the foundation for avoiding undesirable foaming of infant formula. 

## 2. Materials and Methods

### 2.1. Materials

D70, D90, WPC, and medium-heat SMP were obtained from five different manufacturers, respectively. The details about these protein ingredients, including chemical composition and source, are shown in Appendix A. The five samples of D70 ingredients were labeled D70-1, D70-2, D70-3, D70-4, and D70-5, respectively. The same labeling method was used for D90, WPC, and SMP samples. All 20 protein ingredients were dissolved using Milli-Q-treated water to a concentration of 10 mg/mL for the further analysis. ANS (8-Anilino-1-naphthalenesulfonic acid) and DTNB (5,5-dithio-bis2-nitrobenzoic acid) were obtained from Sigma Chemical Co. (St. Louis, MO, USA). The chemicals used in this experiment were of analytical grade.

### 2.2. Characterization of Foaming Properties

#### 2.2.1. Foaming Capacity and Foam Stability

The FC and FS were measured according to the method of Wang et al. [13]. A 50 mL protein solution (10 mg/mL) was placed in a cylindrical glass container (5.5 cm in diameter and 10.5 cm in height) and stirred with a high-speed disperser (Ultra Turrax T25, IKA Labortechnik, Staufen, Germany) at a speed of 10,000 r/min for 2 min. The volumes were measured at 0 and 30 min after stirring (denoted V_0_, V_30_). FC and FS were calculated according to the following formulas:(1)$$\mathrm{FC}\;(\%)=\frac{V_{0}\;-\;50}{V_{0}}\;\times \;100$$
(2)$$\mathrm{FS}\;(\%)=\frac{V_{30}\;-\;50}{V_{0}\;-\;50}\;\times \;100$$

#### 2.2.2. Foam Morphology Analysis

The morphology of foam was observed using an optical microscope (BX53, Olympus, Tokyo, Japan). The foam was prepared as described above (Section 2.2.1) and placed on the glass slide. Foam morphology was observed under ×4 objective lens and ×10 eye lens. The experiment was repeated at least five times.

### 2.3. Particle Size Measurement

The particle size was characterized by dynamic light scattering as described by Zhang et al. [14] with Zetasizer Nano-ZS (Malvern Instruments Ltd., Worcestershire, UK). Solutions of D70, D90, WPC, and SMP were diluted at concentration of 1 mg/mL, and the refractive indices of protein and water were 1.45 and 1.33, respectively. Particle size was expressed as a volume-weighted mean diameter (D43).

### 2.4. Zeta Potential Measurement

Zeta potential was measured by Zetasizer Nano ZS (Malvern Instruments, UK). Samples were dissolved in Milli-Q water to prepare a 1 mg/mL protein solution and measured at 25 °C.

### 2.5. Protein Turbidity and Solubility Measurement

Turbidity was determined according to Zhao et al. [15]. The absorbance of protein solution (2 mg/mL) was measured at 633 nm using an ultraviolet (UV) spectrophotometer (T9, PUXI, Beijing, China).

For solubility, the protein solution was centrifuged at 10,000× *g* for 15 min at room temperature. The supernatant was taken, and the concentration was determined using BCA method (Beyotime, Shanghai, China). Protein solubility was described as the ratio of the protein concentration of supernatant and total protein concentration.

### 2.6. Sodium Dodecyl Sulphate-Polyacrylamide Gel Electrophoresis (SDS-PAGE)

Gel electrophoresis under reducing and non-reducing conditions was performed according to the method of Cui et al. [16]. Briefly, the loading buffer (5×) was mixed with protein solution (2 mg/mL) and then heated in a boiling water bath for 5 min. Mixed samples of 10 µL were loaded on the gels composed of 12% separation gel and 5% concentration gel. The SDS-PAGE was conducted at 120 mV with a molecular weight standard of 10.0–180.0 kDa (Biosharp, Beijing, China). Gels were scanned using a two-color infrared laser imaging system (Odyssey CLX, LICOR, Lincoln, NV, USA). 

### 2.7. Circular Dichroism (CD) Spectroscopy

Structure of the protein sample was studied by circular dichroism spectrometer (Chirascan, Optical Physics Inc, London, NK), according to the method of Pi et al. [17]. The wavelength range and bandwidth were 190–260 nm and 1 nm, respectively. The content of the secondary structure was calculated by online CONTIN program at http://dichroweb.cryst.bbk.ac.uk/html/home (accessed on 17 April 2022).

### 2.8. Surface Hydrophobicity Measurement

Surface hydrophobicity was measured according to the method by Liu et al. [18] using fluorescence spectrophotometer (F-7000, Hitachi Corp., Hitachi City, Japan). ANS (8.0 mmol/L, pH 7.0, 20 µL) was added to 4 mL of sample (0.5 mg/mL). The excitation wavelength was 370 nm, emission wavelength was set to 400–700 nm, emission slits were 5 nm, and PMT potential was 600 V. The maximum fluorescence intensity was used as hydrophobicity index and noted as H0. 

### 2.9. Free Sulfhydryl (SH) Group Content Measurement

Free SH content was measured according to the method proposed by Peng et al. [19]. A protein solution (5 mg/mL) was added to 2.5 mL tris-glycine buffer (0.086 M tris,0.09 M glycine, 0.004 M EDTA, pH 8.0) and 0.05 mL Ellman reagent (40 mg DTNB dissolved in 10 mL Tris-Gly buffer). The absorbance at 412 nm was measured. Free SH content was calculated according to the following formula:(3)$$\mathrm{Free}-\mathrm{SH}\;(\mu \mathrm{mol}/g)=(73.53{\;\times \;\mathrm{OD}}_{412}\text{× }D)\;/\;C$$

OD_412_ is the absorbance of the sample at 412 nm, D is the dilution factor, and C is the protein concentration.

### 2.10. Data Analysis

All experiments were repeated three times; data were expressed as mean ± standard deviation. The data was analyzed by SPSS 20.0 (SPSS, Inc., Chicago, IL, USA) and Origin 2017 (OriginLab Corporation, Northampton, MA, USA). The difference between samples was measured using principal component analysis (PCA). The correlation between foaming properties and physiochemical properties was estimated by Pearson’s correlation analysis.

## 3. Results and Discussion

### 3.1. Foaming Properties

#### 3.1.1. Foaming Capacity (FC) and Foam Stability (FS)

Significant differences (*p* < 0.05) in FC were observed for each protein ingredient from different manufacturers (Figure 1A). In these four protein ingredients, FC values for D70 D90, WPC, and SMP ranged from 69.1% to 95.6%, 65.3% to 87.8%, 63.2 to 85.5%, and 77.9 to 103.4%, respectively. These results showed that the undesirable foam in infant formula protein ingredients varied from different manufacturers. Protein ingredients with low FC, including D70-1, D90-3, WPC-5, and SMP-5, might be appropriate for the production of infant formula. The average FC values of five different manufacturers were calculated to investigate the foaming properties of different types of protein ingredients (Appendix A). SMP showed a highest FC (90.7 ± 9.7%), followed by D70 (80.3 ± 9.7%), D90 (78.8 ± 8.6%), and WPC (74.1 ± 9.7%). This phenomenon probably results from the existence of casein in SMP, which unfolds easily at the air–water interface due to its high flexibility [20]. 

Significant differences (*p* < 0.05) in FS were also found in each protein ingredient from different manufacturers (Figure 1B), indicating that different protein ingredients had a great impact on the undesirable foam of infant formula. In particular, the FS of D70 and D90 from five different manufacturers varied significantly, with the lowest FS of 22.2% and 16.9% and the highest of 65.0% and 46.9%, respectively. In the foam stability of different types of protein ingredients, the highest FS existed in SMP (58.2 ± 5.5%), while that of D70, D90, and WPC were 37.5 ± 18.7%, 31.6 ± 13.0%, and 16.5 ± 4.3%, respectively (Appendix A). It was probably because of the existence of casein in SMP, which could adsorb and diffuse at the interface to form a rigid film that contributes to the stability of the foam [21]. The same results were observed by Xiong et al., who reported that increasing the casein: whey protein ratio from 40:60 to 80:20 resulted in improving the FS of milk protein dispersion [22]. D70 and D90 showed higher FS than WPC, which can probably be attributed to the presence of lactose. According to a Sun et al. study, the addition of saccharides to egg white improved FS by increasing the apparent viscosity and inhibiting foam expulsion [23]. The lower FS was probably due to the high content of fat, which might be disadvantageous for forming a viscoelastic interfacial layer to stabilize the foam [24]. 

#### 3.1.2. Foam Structure

To further evaluate the foaming properties of the samples, the microscopic morphology was observed (Figure 1C). The smallest bubbles were formed in SMP compared to D70, D90, and WPC. Smaller bubbles are more resistant to condensation and drainage, resulting in foam with higher drainage stability [22]. These results were in agreement with those of FC and FS: SMP showed the highest FC and FS. The larger bubbles are more unstable and tend to collapse due to disproportionation, condensation, and drainage [13]. Differences in bubble size and distribution were observed in D70 and D90 from different manufacturers. Among these, D70-1, D70-3, D70-5, D90-4, and D90-5 displayed larger and non-uniform bubbles, which was consistent with the results of FS described above (Figure 1B). Meanwhile, for WPC, no differences in bubble size and distribution state were observed in different manufacturers although it showed large bubble size. 

### 3.2. Particle Size, Zeta Potential, Turbidity, and Solubility

In each type of protein ingredient from different manufacturers, the particle size showed a significant difference (*p* < 0.05). D70-5, D90-1, WPC-5, and SMP-5 showed the highest particle size in comparison to the samples from other manufacturers, which may be related to processing conditions [25]. The large particle size was observed in WPC (280.5 ± 47.9 nm), followed by D70 (261.4 ± 33.9 nm), D90 (227.7 ± 23.5 nm), and SMP (223.6 ± 15.5 nm) (Appendix A). WPC exhibited the largest particle size, mainly resulting from the high content of fat (Appendix A). These results were consistent with the study of Banavara et al., who indicated that fat enhances the surface adhesion of the powder to increase in particle size [12]. The changes in turbidity were similar to the particle size (Figure 2C), which was also reported by Du et al. [26].

Zeta potential results are shown in Figure 2B. Significant differences (*p* < 0.05) in the zeta potential were shown in each type of protein ingredients from different manufacturers. This may be due to the exposure of different negatively charged groups buried in the proteins during processing [27]. The low absolute charge may result in a high FC in D90-1, D90-2, WPC-3, and WPC-4. Notably, the FC is also influenced by particle size and molecular flexibility besides the zeta potential [28]. D70-4, with the highest absolute charge, showed the highest FC among others D70 protein ingredients, which may be influenced by its small particle size. In the four protein ingredients, SMP showed the lowest absolute surface charge (−17.6 mV) compared to D70 (−22.3 mV), D90 (−25.3 mV), and WPC (−23.7 mV) (Appendix A). The low absolute charge could promote foaming properties by reducing the electrostatic potential barrier at the interface, facilitating the rapid adsorption of molecules [29], which may account for the high foaming properties of SMP. 

Differences in solubility were observed among D90 and SMP ingredients from different manufacturers, while there were no significant (*p* < 0.05) differences among the D70 and WPC (Figure 2D). Among the D90 and SMP protein ingredients from different manufacturers, the highest solubility was observed in D90-3 (96.7 ± 3.1%) and SMP-3 (73.6 ± 10.0%), respectively. Le et al. revealed that the exposure of polar groups enhanced protein–water interactions to result in an increase of solubility during processing [30]. There were significant (*p* < 0.05) differences in foaming properties between D70 and WPC ingredients from different manufacturers although they had no significant differences in the solubility. Similar results were reported by Chang et al. [31]. In Appendix A, the solubility of D70, D90, WPC, and SMP were 93.7 ± 3.8%, 88.5 ± 6.9%, 86.4 ± 6.6%, and 65.5 ± 6.1%, respectively. Considering that casein exists as colloidal spherical particles and showed to be less soluble than whey protein [32], it was found that SMP showed the lowest solubility, probably resulting from the existence of casein. Interestingly, SMP exhibited the excellent foaming properties despite its low solubility, which may be attributed to its small particle size and low absolute surface charge. 

### 3.3. Protein Composition

To investigate the protein composition, a reducing SDS-PAGE was conducted (Figure 3). In D70, D90, and WPC ingredients, three major protein bands (18 kDa, 13 kDa, and 66 kDa) were identified as *β*-lactoglobulin (*β*-Lg), *α*-lactalbumin (*α*-La), and lactoferrin (Lf), respectively. Three major bands were observed in SMP ingredients at around 13 kDa, 18 kDa, and 35 kDa, which were considered *α*-La, *β*-Lg, and casein, respectively. These results were in agreement with Jiang and Rupp et al. [33,34]. For the same type of ingredients, no visible differences were identified under reducing condition, indicating a similar protein composition from different manufacturers’ proteins. 

As shown in Figure 3, a non-reducing SDS-PAGE was performed. Different aggregates (>180 kDa) were observed in each type of ingredient under non-reducing conditions. These aggregates disappeared after reduction, suggesting that these aggregates were linked via disulfide bonds. Heat processing during pasteurization and spray drying may be a reasonable explanation for protein aggregation [35]. Similar results were shown in Gazi et al., where casein and *β*-Lg were found to aggregates during preheating, concentration, and spray drying [36]. The aggregations could affect the foaming properties. Protein aggregates may contribute to the stability of foam by increasing the viscoelasticity of the air–water film [37,38]. On the other hand, the large size and compact structure of aggregates may reduce the stability of foam [39].

### 3.4. Circular Dichroism (CD) Spectra

CD spectra and secondary structure content (*α*-helix, *β*-sheet, *β*-turn, and random coil) of these four protein ingredients are shown in Figure 4A,B. D70, D90, and WPC ingredients displayed similar CD profiles for their similar protein composition, which were consistent with the SDS-PAGE results (Figure 3). However, a negative peak around 200 nm with random coil structure was observed in SMP ingredients. This may result from the presence of caseins in SMP, which contains 20% *α*-helix and 60–80% random coil structure [40]. Variations in CD band intensity were found in protein ingredients from different types and manufacturers. The changes at 208 nm among all 20 ingredients indicated differences in *α*-helix content. As shown in Appendix A, significant (*p* < 0.05) differences in secondary structure content were found in SMP compared to other types of protein ingredients. Low contents of *α*-helix (14.8%) and *β*-sheet (25.5%) were found in SMP compared to D70 (15.9% for *α*-helix; 28.2% for *β*-sheet), D90 (16.2% for *α*-helix; 27.9% for *β*-sheet), and WPC (15.6% for *α*-helix; 29.2% for *β*-sheet), while high content of *β*-turn (24.9%) and random coil (34.6%) existed in SMP. Differences in secondary structure content affect the foaming properties. The high content of random structure found in SMP indicated its higher flexibility, and the flexible molecular structure contributes to improve the foaming properties [41].

### 3.5. Surface Hydrophobicity (H0) and Free Sulfhydryl (SH) Content

Surface hydrophobicity (H0) is an indicator reflecting the conformational of protein (Figure 5A). Differences were also found in each type of ingredient from different manufacturers. Molecules unfolding and exposure of hydrophobic amino acid residues happened during processing, resulting in an increase of H0 values [26], while protein aggregation further reduces the protein surface area and the binding of ANS to the hydrophobic point, resulting in a decrease of H0 values. The increase of H0 improves the foaming properties due to the rapid penetration of the protein into the air phase as well as facilitating the rearrangement and interaction of the protein molecules [42]. In D70, D90, and WPC ingredients, the change of H0 values had a positive correlation to those of FC. Differently, the H0 values of SMP showed no relationship with FC (Figure 1A). These results were consistent with Zhan et al., who reported that the H0 values of sodium caseinate decreased, and an increase of FC was also shown under treatment with gallic acid [43]. The four types ingredients were at the order of D70 (5014.6 ± 556.6) > D90 (4667.6 ± 702.7) > WPC (3138.4 ± 186.4) > SMP (2999.5 ± 245.6) (Appendix A). Differences in H0 and SH content reflect changes in protein conformational structure. The SH content varied from the protein ingredient types and manufacturers (Figure 5B). D70, D90, and WPC contained high SH content compared to SMP (Appendix A), which may be due to the high content of SH proteins in D70, D90, and WPC, such as *β*-Lg and BSA. 

### 3.6. Principal Component and Correlation Analysis

PCA showed the distribution of different protein ingredients in a multidimensional space based on all their properties (Figure 6A). PC1 and PC2 accounted for 51.5% and 20.3%, respectively. A slight scatter phenomenon occurred in the same type of ingredients from different manufacturers, indicating that the ingredients were different according to their origin. SMP was concentrated in the third quadrant, while D70 and D90 were mainly located in the first and fourth quadrants, and WPC was located in the fourth quadrant, indicating differences among the four types of protein ingredients. 

The correlation between foaming and physicochemical properties was further analyzed (Figure 6B). *β*-strand showed a negative correlation with FC/FS (−0.58/−0.71), while *β*-turn and random coil exhibited a positive correlation (0.53/0.73 for the *β*-turn and 0.55/0.74 for the random coil). SH content was negatively correlated with foaming properties, which was consistent with Zhang et al., who reported that the improvement in foaming properties was due to the masking of SH on the protein surface [44]. Zeta potential was positively correlated with FC/FS (0.55 and 0.51). The decrease in zeta potential resulted in the rapid lessening of the electrostatic barrier at the interface and the promotion of protein adsorption at the air–water interface [24], thus improving the foaming properties. Particle size was negatively correlated with FC and FS (−0.45 and −0.53), suggesting that particles with smaller size will be beneficial in improving foam properties [13]. Moreover, the negative correlation between fat content and FC/FS (−0.5/−0.49) implied that fat is not capable of forming the viscoelastic interfacial layer required to stabilize the foam bubbles. The lactose content and H0 showed little influence on foaming properties.

Based on the aforementioned results, the foaming properties of protein ingredients are determined by several factors, such as composition (protein, fat, and lactose), conformation (secondary and tertiary structure), and surface properties (surface charge and surface hydrophobicity). Interestingly, conformation showed a more significant relationship with foaming properties than with composition (e.g., fat and lactose). Both processing and raw materials can change the composition and physicochemical properties, which can affect the foaming properties of protein ingredients. The high content of disordered structure results in improved foaming of the protein ingredients, which may be explained by the structural unfolding caused by the heat treatment during processing [9].

## 4. Conclusions

The foaming and physicochemical properties of D70, D90, WPC80, and SMP were characterized and compared in this study. Significant differences in foaming properties (*p* < 0.05) occurred not only among types of protein ingredients but also among same ingredients from different manufacturers, which was related to the composition and physicochemical properties in protein. The foaming properties showed stronger relationships with secondary structure, especially *β*-strand, *β*-turn, and random coil, compared to the composition. From the results we found that a high content of *β*-strand, SH content, absolute zeta potential, particle size, fat content, and low content of *β*-turn and random coil were statistically associated with low foaming. Overall, the data of this study provided foaming information of the protein ingredients, which could supply a strategy for the development of a low-foaming infant formula formulation.

## Figures and Tables

**Figure 1 foods-11-03710-f001:**
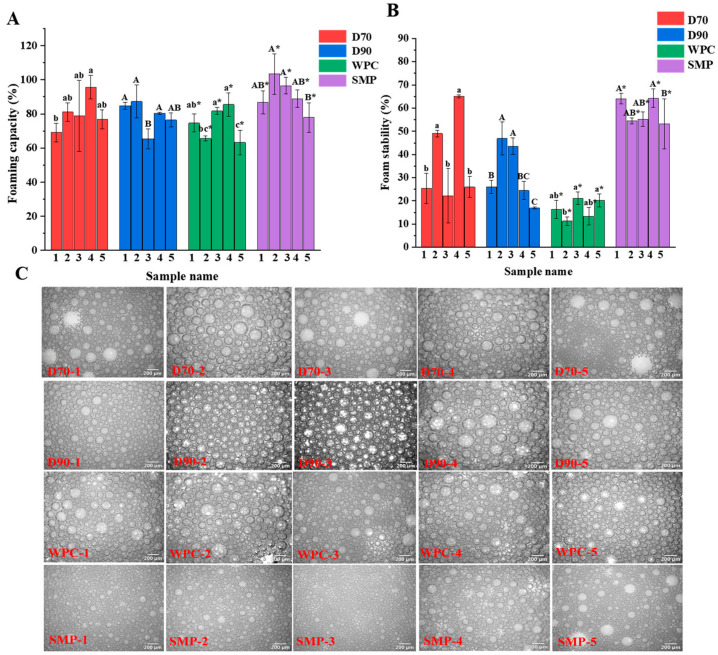
Foaming properties of D70, D90, WPC, and SMP samples from different manufacturers. Foaming capacity (**A**), foam stability (**B**), and micrograph of bubble under microscope (**C**). Different letters (a–b, A–C, a*–c*, A*–B*) indicate significant differences at *p* < 0.05.

**Figure 2 foods-11-03710-f002:**
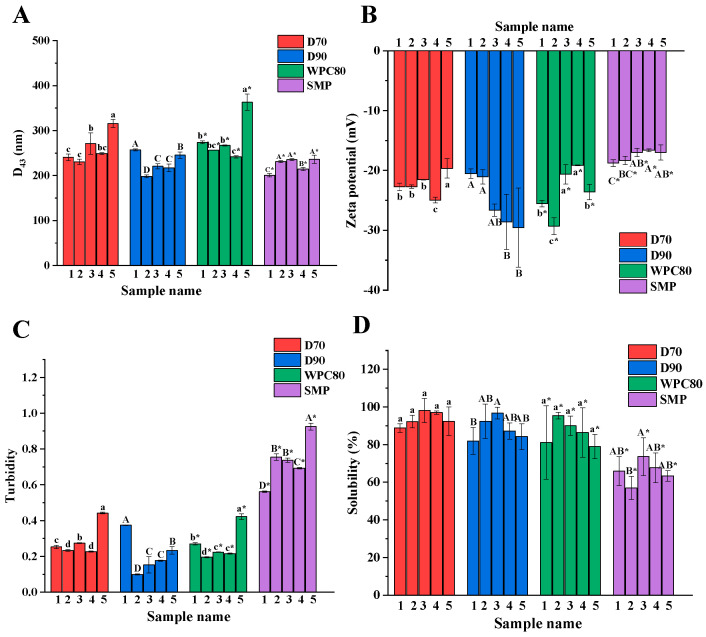
Physicochemical properties of D70, D90, WPC, and SMP samples from different manufacturers. Particle size (**A**), zeta potential (**B**), turbidity (**C**), and solubility (**D**). Different letters (a–d, A–D, a*–d*, A*–D*) indicate significant differences at *p* < 0.05.

**Figure 3 foods-11-03710-f003:**
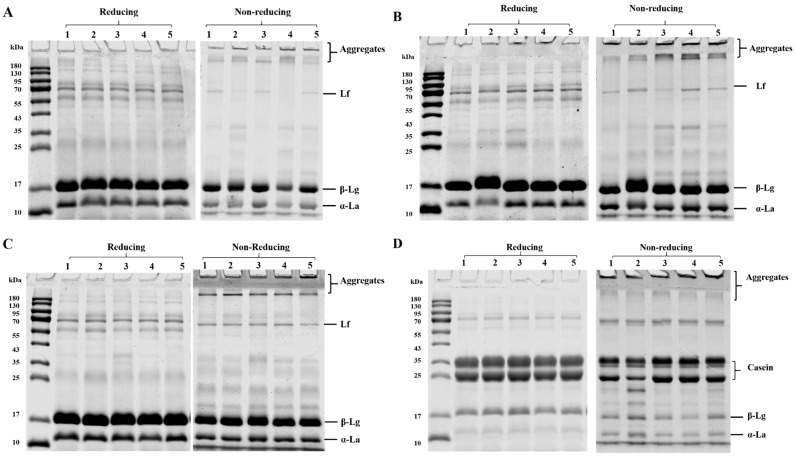
Sodium dodecyl sulphate–polyacrylamide gel electrophoretograms (SDS–PAGE) under reducing and no-reducing conditions of D70 (**A**), D90 (**B**), WPC (**C**), and SMP (**D**).

**Figure 4 foods-11-03710-f004:**
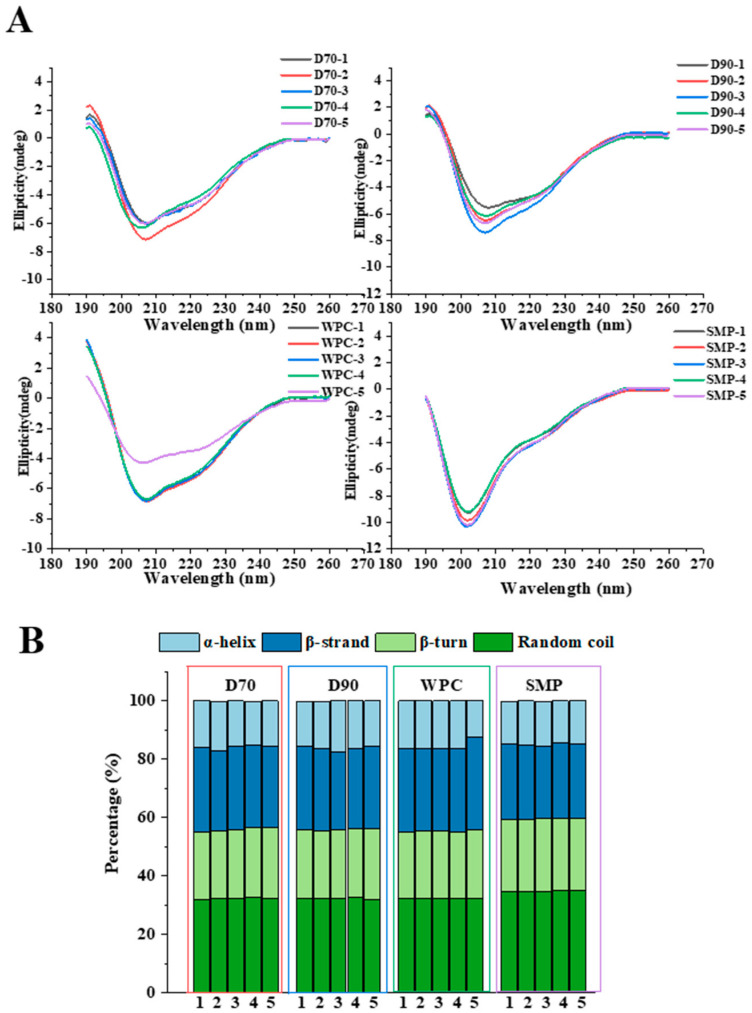
Circular dichroism spectroscopy of D70, D90, WPC, and SMP samples. Circular dichroism spectroscopy (**A**); secondary structure content (**B**).

**Figure 5 foods-11-03710-f005:**
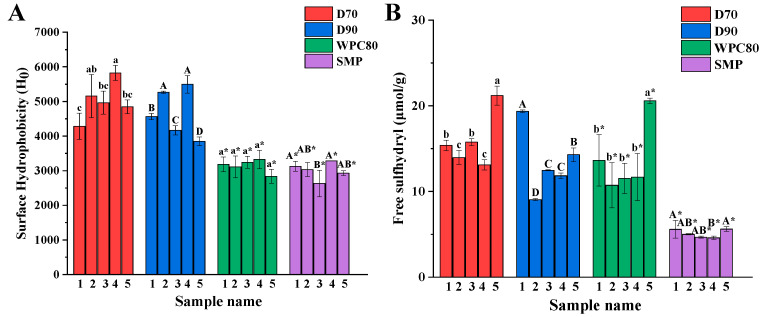
Surface hydrophobicity (**A**) and free sulfhydryl content (**B**) of D70, D90, WPC, and SMP samples from different manufacturers. (a–c, A–D, a*–b*, A*–B*) indicate significant differences at *p* < 0.05.

**Figure 6 foods-11-03710-f006:**
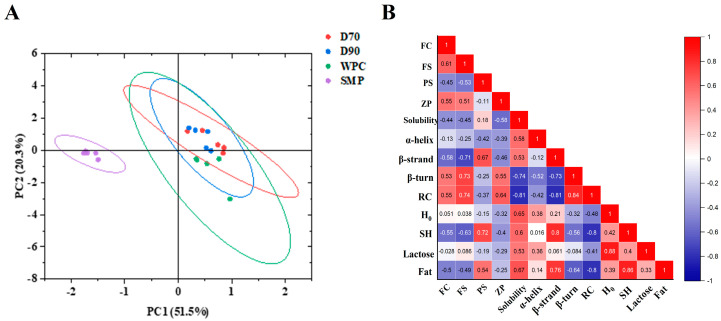
Principal component analysis score plot (**A**) and correlation analysis result (**B**) for D70, D90, WPC, and SMP samples. FC, foaming capacity (%); FS, foam stability (%); PS, particle size (nm); RC: random coil (%); H0, surface hydrophobicity; SH, free sulfhydryl content (μmol).

## Data Availability

The datasets generated for this study are available on request to the corresponding author.

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
