# Peer review of "Foaming and Physicochemical Properties of Commercial Protein Ingredients Used for Infant Formula Formulation"

_foods, 2022, doi:10.3390/foods11223710_

Round 1

Reviewer 1 Report

In this study, entitled "Foaming and physicochemical properties of commercial protein ingredients used for infant formula formulation", the authors mainly aim at finding a link between protein physico-chemical properties and foaming dynamics occurring during infant formula reconstitution. Indeed, this topic has an original scientific interest and at the same time an evident industrial application. However, the presentation of the strategy and the results, the lack of deep interpretation, and the multiple imperfections in terms of English grammar and sentence structure.

Here below, I provide some insights for the authors.

In the introduction, beyond a lack of information about the samples investigated and their characteristics, it is not clear what the authors are going to explore and how, since the open question is quite vast. In the Materials and Methods section, the different protocols are not always explained, citing mainly previous works. Therefore, it is difficult to understand how to interpret the results. Mostly, despite testing different types (from different manufacturers) of different protein powders, the authors sometimes present average results (for example Table1) and some others they highlight the difference between the samples. This is quite confusing and the possible interpretation of the outcomes is quite vague. Moreover, a real characterization of the composition of the powders is not provided, thus making difficult to link the properties at the micro-scale to the foaming behavior. Indeed, in the conclusion, the authors say that the differences between different protein products "may be" related to the physico-chemical properties of the proteins, thus making this work quite approximate.

Author Response

Reviewers' comments:

Reviewer #1:

In this study, entitled "Foaming and physicochemical properties of commercial protein ingredients used for infant formula formulation", the authors mainly aim at finding a link between protein physico-chemical properties and foaming dynamics occurring during infant formula reconstitution. Indeed, this topic has an original scientific interest and at the same time an evident industrial application. However, the presentation of the strategy and the results, the lack of deep interpretation, and the multiple imperfections in terms of English grammar and sentence structure.

Here below, I provide some insights for the authors.

  1. In the introduction, beyond a lack of information about the samples investigated and their characteristics, it is not clear what the authors are going to explore and how, since the open question is quite vast.
  2. In the Materials and Methods section, the different protocols are not always explained, citing mainly previous works. Therefore, it is difficult to understand how to interpret the results.
  3. Mostly, despite testing different types (from different manufacturers) of different protein powders, the authors sometimes present average results (for example Table1) and some others they highlight the difference between the samples. This is quite confusing and the possible interpretation of the outcomes is quite vague.
  4. Moreover, a real characterization of the composition of the powders is not provided, thus making difficult to link the properties at the micro-scale to the foaming behavior.
  5. Indeed, in the conclusion, the authors say that the differences between different protein products "may be" related to the physico-chemical properties of the proteins, thus making this work quite approximate.

Authors: Thank you for your valuable comments. The English grammar and sentence structure has been checked carefully, please see the revised text.

  1. In the introduction, the information about samples investigated and their characteristics were added, and the exploration content and exploration method were clarified. Please see Lines 47-82, Page 2 in the revised text highlighted in yellow.
  2. We have revised the Materials and Methods section, please see Lines 83-162, Page 2-4 in the revised text highlighted in yellow.
  3. The results have been focused on differences among the same protein ingredients from different manufactures. Differences among different types of ingredients has been also revised. In addition, more discussion has been added in the text. Please see Lines 166-186, Page 4; Lines 211-216, Page 5; Lines 221-230, 233-238, 241-244, Page 6; 296-304, Page 8-9. Lines 339-347, Page 9-10 in the revised text highlighted in yellow.
  4. The information of protein ingredients has been uploaded as the supplementary files. For a better view, we have also listed the information at the text. Please see Table S1 Line 472, Page 13.
  5. The conclusion has been revised carefully, please see Lines 353-363, Page 10 in the revised text highlighted in yellow.

Lines 353-363, Page 10. “The foaming and physicochemical properties of D70, D90, WPC, and SMP were characterized and compared in this study. Significant differences (p < 0.05) occurred not only among types of protein ingredients, but also among same ingredients from different manufacturers. The composition and physicochemical properties were responsible for these differences. The foaming properties showed stronger relationships with secondary structure especially β-strand, β-turn, and random coil compared to composition. It can be assumed that protein ingredients with high content of β-strand, SH content, absolute zeta potential, particle size, fat content and low content of β-turn and random coil were more suitable for low-foaming infant formula formulation. Overall, data of this study provided foaming information of protein ingredients, which could supply a strategy for the development of low-foaming infant formula formulation”.

Reviewer 2 Report

In the research work, the foaming and physicochemical properties of four protein components from different manufacturers were analyzed. The conclusions of the conducted research are clear and result from the obtained research results. The material used for the research is sufficient, the research methods have been selected appropriately. Discussing the results against the background of other authors is very detailed. The publications cited by the authors of the article are well selected. For the most part, the authors refer to the latest knowledge published in renowned scientific journals. I could not find any mistakes in the scientific aspect of the manuscript.

However, the authors did not avoid a few mistakes, which I will list below:

- A few punctuation problems are present in the manuscript. I suggest the Authors to double-check the text.

-The arrangement of the figure is not clearly legible, it suggests a better display of the letters showing the statistical differences between the samples (Tables 1, 2 and 5).

Author Response

Reviewers' comments:

Reviewer #2:

  1. A few punctuation problems are present in the manuscript. I suggest the Authors to double-check the text.

Authors: Thanks for your valuable comments.

We have checked the punctuation problems for the whole text carefully. Please see Line 112, 114, Page 3; Line 187, Page 4; Line 256, Lines 306-308, Page 9 in the revised text highlighted in yellow.

  1. The arrangement of the figure is not clearly legible, it suggests a better display of the letters showing the statistical differences between the samples (Tables 1, 2 and 5).

Authors: Thank you for your valuable comments.

We have arranged the figure clearly to show the letters of statistical differences between samples. Please see Figure 1, Line 194, Page 5; Figure 2, Line 247, Page 6; Figure 5, Line 313, Page 9, respectively.

Reviewer 3 Report

Article is well written and provides detailed information on the structure and composition of commercial infant formula. I would suggest authors formulate the conclusion/discussion a bit more and report from their results what would be for instance the most relevant parameter to control when trying to avoid foaming. In the conclusion they listed many of them (i.e., high fat content, large particle size, high absolute zeta potential, and high con-343 tent of β-strand and SH), which one is more relevant? Is it more related to the compositional variables of the product as they have analyzed or with the preparation conditions?

Some other comments:

Line: One of the main ingredients for what? Specify.

Line 42 and 45: Change “during which”

Line 82: Provide details on how the 50 mL protein solution (10 mg/mL) was done. Were the samples diluted in hot/cold water?

Line 103: fix degree sign

Line 109: At which temperature?

Line 149: Re-write

Table 1: Insert it after the discussion section. For these results an average of all 5 manufactures were considered? Do author consider averaging different samples viable? Wouldn’t it be more valuable to carry out the difference test among same protein from different manufactures? Maybe authors can use Table 1 as a general discussion for all analysis in the end of discussion section instead of in the beginning.

Line 193: in agreement

Line 345: data

Author Response

Reviewers' comments:

Reviewer #3:

  1. I would suggest authors formulate the conclusion/discussion a bit more and report from their results what would be for instance the most relevant parameter to control when trying to avoid foaming. In the conclusion they listed many of them (i.e., high fat content, large particle size, high absolute zeta potential, and high content of β-strand and SH), which one is more relevant? Is it more related to the compositional variables of the product as they have analyzed or with the preparation conditions?

Authors: Thank you for your valuable comments.

The composition and physicochemical properties are the two main factors for foaming. Pearson's correlation analysis showed that the foaming properties showed stronger relationships with secondary structure especially β-strand, β-turn, and random coil compared to composition. Among the parameters listed in conclusion, the content of β-strand, β-turn, and random coil were more relevant than others. More discussion has been added, please see Lines 339-347, Page 9-10. The conclusion has also been rewritten. Lines 353-363, Page10 in the revised text highlighted in yellow.

Lines 339-347, Page 9-10. “Based on the aforementioned results, the foaming properties of protein ingredients are determined by several factors, such as composition (protein, fat, and lactose), con-formation (secondary and tertiary structure), and surface properties (surface charge and surface hydrophobicity). Interestingly, conformation showed more relationship with foaming properties rather than composition (i,e., fat, lactose). Both processing and raw materials can change the structural and foaming properties of these ingredients. The increase in disorder structure could result in an increasing foaming of the protein ingredients, which may be explained by the structural unfolding caused by the heat treatment during processing [9].”

353-363, Page10. “The foaming and physicochemical properties of D70, D90, WPC80, and SMP were characterized and compared in this study. Significant differences (p < 0.05) occurred not only among types of protein ingredients, but also among same ingredients from different manufacturers. The composition and physicochemical properties were responsible for these differences. The foaming properties showed stronger relationships with secondary structure especially β-strand, β-turn, and random coil compared to composition. It can be assumed that protein ingredients with high content of β-strand, SH content, absolute zeta potential, particle size, fat content and low content of β-turn and random coil were more suitable for low-foaming infant formula formulation. Overall, data of this study provided foaming information of protein ingredients, which could supply a strategy for the development of low-foaming infant formula formulation.”

  1. Line: One of the main ingredients for what? Specify.

Authors: Thank you for your valuable comments.

We have specified in the manuscript, please see Lines 11-12, Page 1 in the revised text highlighted in yellow.

Lines 11-12, Page 1 “Protein, as one of the main ingredients for infant formula, may be closely related to the undesirable foam formed during the reconstitution of infant formula”.

  1. Line 42 and 45: Change “during which”

Authors: Thank you for your valuable comments.

We have change “during which” to “among them”, please see Line 44, Page1 in the revised text highlighted in yellow.

Lines 44, Page1. “Among them, D70 and D90 ...”

  1. Line 82: Provide details on how the 50 mL protein solution (10 mg/mL) was done. Were the samples diluted in hot/cold water?

Authors: Thank you for your valuable comments.

We have supplemented the details in text, please see Lines 92-93, Page 2 in the revised text highlighted in yellow.

Lines 92-93, Page 2. “All 20 protein ingredients were dissolved using Milli-Q treated water to a concentration of 10 mg/mL for the further analysis.”

  1. Line 103: fix degree sign

Authors: Thank you for your valuable comments.

We have revised this part according to your suggestion. Please see Line 118, Page 3 in the revised text highlighted in yellow.

Line 118, Page 3. “Samples … at 25oC.”

  1. Line 109: At which temperature?

Authors: Thank you for your valuable comments.

We have supplemented the information, please see Lines 123-124, Page 3 in the revised text highlighted in yellow.

Lines 123-124, Page 3. “For solubility, …at room temperature...”

  1. Line 149: Re-write

Authors: Thank you for your valuable comments.

We have revised this part according to your suggestion. Please see Lines 158-162, Page 4 in the revised text highlighted in yellow.

Lines 158-162, Page 4. “All experiments were repeated three times, data were expressed as mean ± stand-ard deviation. The data was analyzed by SPSS 20.0 (SPSS, Inc., Chicago, IL, USA) and Origin 2017 (OriginLab Corporation, Northampton, USA). The difference between the samples was measured using Principal component analysis (PCA). The correlation between foaming properties and other physiochemical properties was estimated by Pearson's correlation analysis.”

  1. Table 1: Insert it after the discussion section. For these results an average of all 5 manufactures were considered? Do author consider averaging different samples viable? Wouldn’t it be more valuable to carry out the difference test among same protein from different manufactures? Maybe authors can use Table 1 as a general discussion for all analysis in the end of discussion section instead of in the beginning.

Authors: Thank you for your valuable comments.

The results have been focused on differences among the same protein ingredients from different manufactures. Differences among different types of ingredients has been also revised. In addition, more discussion has been added in the text. Please see Lines 166-186, Page 4; Lines 211-216, Page 5; Lines 221-230, 233-238, 241-244, Page 6; 296-304, Page 8-9. Lines 339-347, Page 9-10 in the revised text highlighted in yellow.

  1. Line 193: in agreement

Authors: Thank you for your valuable comments.

We have revised this part according to your suggestion. Please see Line 202, Page 5 in the revised text highlighted in yellow.

Line 202, Page 5. “This result was in agreement …”

  1. Line 345: data

Authors: Thank you for your valuable comments.

We have revised this word according to your suggestion. Please see Line 361, Page 10 in the revised text highlighted in yellow.

Line 361, Page 10. “Overall, data of…”

Round 2

Reviewer 3 Report

Authors have made appropriate changes throughout the manuscript.

Author Response

Authors: Thank you for your valuable comments. The English grammar and sentence structure has been checked carefully, please see the revised text.